# Three-dimensional imaging of waves and floes in the marginal ice zone during a cyclone

Alberto Alberello [1,2,14] ✉, Luke G. Bennetts [1], Miguel Onorato[3,4], Marcello Vichi [5,6], Keith MacHutchon[7], Clare Eayrs [8], Butteur Ntamba Ntamba [9], Alvise Benetazzo[10], Filippo Bergamasco[11], Filippo Nelli[12], Rohinee Pattani[13], Hans Clarke[2], Ippolita Tersigni[2] & Alessandro Toffoli [2]

The marginal ice zone is the dynamic interface between the open ocean and consolidated inner pack ice. Surface gravity waves regulate marginal ice zone extent and properties, and, hence, atmosphere-ocean fluxes and ice advance/retreat. Over the past decade, seminal experimental campaigns have generated much needed measurements of wave evolution in the marginal ice zone, which, notwithstanding the prominent knowledge gaps that remain, are underpinning major advances in understanding the region's role in the climate system. Here, we report three-dimensional imaging of waves from a moving vessel and simultaneous imaging of floe sizes, with the potential to enhance the marginal ice zone database substantially. The images give the direction–frequency wave spectrum, which we combine with concurrent measurements of wind speeds and reanalysis products to reveal the complex multi-component wind-plus-swell nature of a cyclone-driven wave field, and quantify evolution of large-amplitude waves in sea ice.

Improved observational capabilities are needed to understand the often paradoxical and baffling regional and inter-annual variabilities of Antarctic sea ice[1,2]. Autonomous platforms that operate in harsh polar environments, such as autonomous underwater vehicles[3] and drones[4], are pushing the boundaries for in-situ observations, generating data for essential calibration and validation of satellite remote sensing, and measuring properties beyond the capabilities of contemporary satellites. The marginal ice zone (MIZ), which is characterised by dynamic interactions between large-amplitude surface waves and relatively small and thin ice floes, is difficult for satellites to capture[5,6] and a

major target for improved observations[7,8]. Wave evolution and ice properties in the MIZ are intimately coupled[9–11], and, hence, there is demand for a technology capable of simultaneously monitoring both wave activity and ice cover properties, which can capture data during storms when wave–ice interactions are most intense.

Historical in-situ measurements of waves in the MIZ show the ice cover attenuates wave energy exponentially over distance[12] at a frequency-dependent rate that induces a downshift of the peak frequency[13], as well as modifying the directional wave spectrum[14]. The attenuation rate has become a research focus, as it informs

[1]University of Adelaide, 5005 Adelaide, SA, Australia. [2]The University of Melbourne, 3010 Parkville, VIC, Australia. [3]Università di Torino, 10125 Torino, Italy. [4]INFN, 10125 Torino, Italy. [5]Department of Oceanography, University of Cape Town, 7701 Rondebosch, South Africa. [6]Marine and Antarctic Research centre for Innovation and Sustainability (MARIS), University of Cape Town, 7701 Rondebosch, South Africa. [7]Department of Civil Engineering, University of Cape Town, 7701 Rondebosch, South Africa. [8]New York University Abu Dhabi, Abu Dhabi, United Arab Emirates. [9]Cape Peninsula University of Technology, 7535 Cape Town, South Africa. [10]Istituto di Scienze Marine, Consiglio Nazionale delle Ricerche, 30122 Venice, Italy. [11]Università Ca' Foscari, 30123 Venice, Italy. [12]Swinburne University of Technology, 3022 Hawthorn, Australia. [13]Atkins, SW1E 5BY London, United Kingdom. [14]Present address: University of East Anglia, NR4 7TJ Norwich, UK. ✉e-mail: alberto.alberello@outlook.com

predictions of the width of the ice-covered region impacted by waves and, hence, the MIZ extent[15,16]. Major advances in measuring wave attenuation in the MIZ have been made over the past decade, including dedicated campaigns in the Arctic[17] and Antarctic[11,18]. State-of-the-art in-situ measurements mostly come from arrays of wave buoys, where the buoys can be traditional open water buoys[19,20] deployed between floes in regions of low ice concentration (usually close to the ice edge) or bespoke buoys deployed on ice floes[11,18,20] large enough to support the buoys (usually away from the ice edge) but small enough that the floes follow the waves. Attenuation rates are generally calculated by applying an exponential decay ansatz to measurements provided by neighbouring buoys, in terms of the significant wave height[11,18,20] or a more detailed analysis in which the ansatz is applied to each component of the one-dimensional (frequency) wave spectrum, under the assumption of a stationary wave field[20–23].

The recent surge in measurements (including remote sensing[24,25]) has generated a new understanding of wave attenuation in the MIZ, particularly on how the wave attenuation rate depends on frequency[21,26]. Certain theoretical models reproduce the observed frequency dependence, but the dominant sources of attenuation are still hotly debated[27] and empirical models often rely upon[15,28]. Further, the measurements have revealed a large range of attenuation rates[23,25], even at comparable frequencies, which is attributed to dependence on ice cover properties, such as ice thickness, areal ice concentration and floe sizes, as well as momentum transfer from winds over the ice cover. Satellite and model-hindcast data have been used to derive empirical relationships between measured attenuation rates and ice concentration[18,22,23], ice thickness[29] and winds[22,23]. In contrast, floe sizes in the MIZ are below satellite resolutions and have only recently been integrated into large-scale models[16,30], so that coincident floe size data have been limited to visual observations during deployment. Overall, data on ice properties are too sparse or unreliable to validate theoretical models.

Stereo-imaging techniques are emerging as a tool for in-situ monitoring of waves and ice properties in the MIZ. In principle, the images can be used from a moving vessel, as in open waters, to reconstruct the sea surface elevation in time and space, thus enabling analysis of wave dynamics in two-dimensional physical space, the frequency–direction spectral domain, and wave statistics[31]. Airborne synthetic aperture radar (SAR) is an alternative method to measure frequency–direction wave spectra and has been applied over 60–80km long transects of the MIZ[32,33]. However, stereo-imaging, being an in-situ technique, can be used to measure sea-ice geometrical properties simultaneously[34] and can be combined with co-located meteorological measurements, e.g., wind velocities. Further, in contrast to SAR[35], stereo-imaging resolves wind sea components of the wave spectrum (short wavelength systems under the influence of local winds), as well as swell (long wavelength systems no longer under the effect of winds).

To date, the use of stereo-imaging techniques in the MIZ has been limited in scope. Campbell et al.[36] use a camera system on a fixed platform on the edge of a lake to quantify incoming and reflected energy fluxes of relatively small waves (<0.3m) in pancake and brash ice. Smith & Thomson[37] use camera images from a moored vessel in the Arctic MIZ during calm conditions (significant wave heights typically around 1m) to calculate bulk wave properties and pancake floe velocities. Alberello et al.[34] use an autonomous stereo-camera system on a vessel moving through the winter Antarctic MIZ during a cyclone to measure pancake floe shapes and sizes.

In this article, we demonstrate the potential to monitor the evolution of the frequency–direction wave spectrum from the images captured by Alberello et al.[34] combined with automated image reconstruction software. We report the extreme sea state created by the cyclone over a >40km transect into the Antarctic MIZ, and validate a subset of the results with co-located buoy measurements. The sea state deep into the MIZ during the cyclone is shown to be more complex than previously thought, and partitioning of the two-dimensional spectra is required to analyse wave evolution of the cyclone-driven wind sea. Further, evidence is given of momentum transfer from winds through 100% ice concentration, based on comparing attenuation of the significant wave height over distance with an empirical model[18], which is considered to be a benchmark due to the large size of the underlying data set and that the measurements were made in the Antarctic MIZ during the sea-ice growth period.

## Results

### Experimental conditions

On the 4th July 2017, the South African icebreaker S.A. Agulhas II entered the Antarctic MIZ at 62° South and 30° East during an explosive polar cyclone. It encountered the ice edge (the northernmost location where ice concentration exceeds 10% in a 1km radius around the vessel[38]) at 08:00 UTC, and reached 100% ice concentration at 09:00 UTC, approximately 10km from the ice edge. It continued South, whilst remaining in 100% ice concentration[34]. Over this time, strong winds (18–19ms$^{-1}$ from North-East according to ERA5 reanalysis, which underestimates in-situ measurements[39]; Fig. 1a) generated extreme sea states in the surrounding area, with significant wave heights ($H_S = 4\sigma_\eta$, where $\sigma_\eta$ is the surface elevation standard deviation) up to 10m North-East of the icebreaker and >6m at the ice edge (i.e., in the 90th

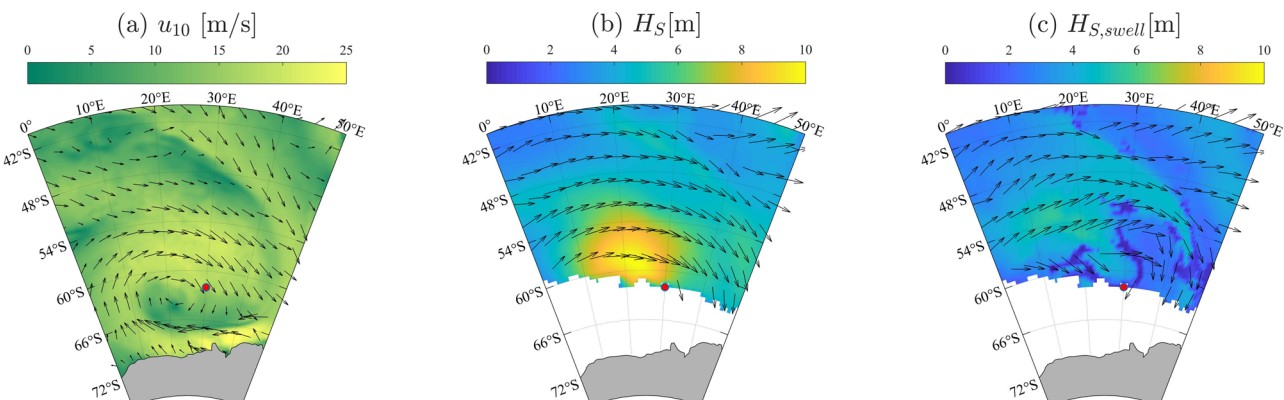

**Fig. 1 | Daily averaged environmental conditions on the 4th July 2017 from ERA5. a** Wind speed; **b** total wave height and **c** swell wave height. Vectors show the direction, where in **b**, **c** the length is proportional to the wave period. The horizontal and vertical axis denotes Easting (longitude) and Northing (latitude) in degrees. **b**, **c** The white area indicates sea-ice concentration ≥15% (ERA5 wave data are only provided for ice concentration up to 15%). The ship position at the ice edge is denoted by the red dot.

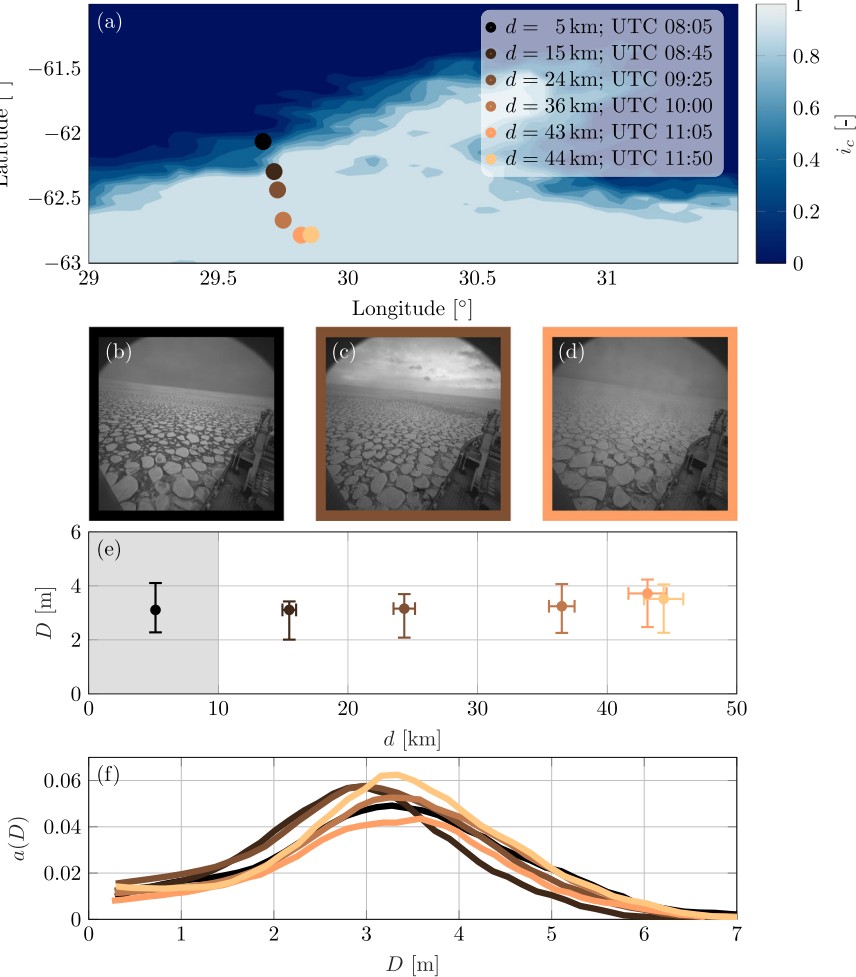

**Fig. 2 | Sea-ice properties on the 4th July 2017. a** Overview of study area with sea-ice concentration from AMSR2 on the 4th July 2017 (longitude and latitude are horizontal and vertical axes, respectively), with bullets indicating six mean measurement locations during 20min acquisitions. **b**–**d** Example of image acquisitions (axes in pixels) at $d=5$km, 24km and 43km from the ice edge, respectively. **e** Median pancake floe diameter ($D$, vertical axis) versus distance from the edge ($d$, horizontal axis) shown as bullets, plus 25th and 75th diameter percentiles (vertical error bars) and uncertainty in distance from ice edge (horizontal error bars). Shaded background denote the measurement location in intermediate ice concentration. **f** Area weighted floe size distribution ($a$, vertical axis) as a function of floe diameter ($D$, horizontal axis) at each measurement location. Colour coding is used in all panels to denote the distance from ice edge.

percentile[40]) when the icebreaker entered the MIZ (Fig. 1b). The mean wave direction at the ice edge was aligned with the wind throughout the day[41] (from North-East). The mean directional spread, a measure of the breadth of the wave spectrum in direction[42], was ≈60° and, like the mean wave direction, was steady during the day[41]. Wave height and period at the ice edge increased throughout the day, due to the intensification of the cyclone, thus creating a non-stationary incident wave field. A detailed analysis of the sea state indicates that ≈70% of the total significant wave height is due to waves generated locally (wind sea). Swell contributes the remaining 30%, which also comes from a north-easterly direction but with a slight offset of less than 20° from the wind sea.

As the icebreaker travelled South into the MIZ, six sequences of three-dimensional images of the ocean surface were captured by a pair of synchronised cameras installed on the icebreaker. The measurement locations are defined by average distances from the ice edge, $d=5$–44km, where the distance is taken along the mean wave direction (Fig. 2a). Each sequence was taken over a 20-minute interval, during which the ship heading and forward speed was almost constant, with the first sequence starting at 08:05 UTC and the last at 11:50 UTC. The 20-minute time interval is a World Meteorological Organisation standard for analysis of wave measurements[43], which balances stationarity

of wave conditions with collecting a large enough number of waves for a statistically robust analysis.

An automatic algorithm for floe size reconstruction was applied to the digital images collected along the transect (see Methods)[34], and used to calculate floe size distributions, mean floe diameters and the areal concentration of floes. At the first measurement location, close to the ice edge ($d=5$km), the ice concentration was $i_c \approx 50\%$[44] and consisted of pancake ice floes (small, approximately circular floes that form in wavy conditions[45,46]) and the remaining 50% was water between the floes (Fig. 2b). At all five subsequent locations, which were deeper into the MIZ ($d \geq 15$km; Fig. 2c, d), ice covered 100% of the ocean surface, in the form of ≈60% pancake floes and ≈40% interstitial frazil ice[34], which increased in density with distance from the ice edge. Pancake floe diameters generally increased with distance from the ice edge, with the median diameter increasing from ≈3.0m at the ice edge to ≈3.5m at the deepest measurement locations (Fig. 2e), and the floe size distribution skewing towards larger diameters (Fig. 2f). At all measurement locations, over 50% of the pancake-covered area was comprised of floes with diameters in the interval 2–4m (vertical error bars in Fig. 2e). Therefore, the ice conditions during the experiment are considered relatively insensitive to distance from the ice edge, in comparison to the changes in the incident wave field.

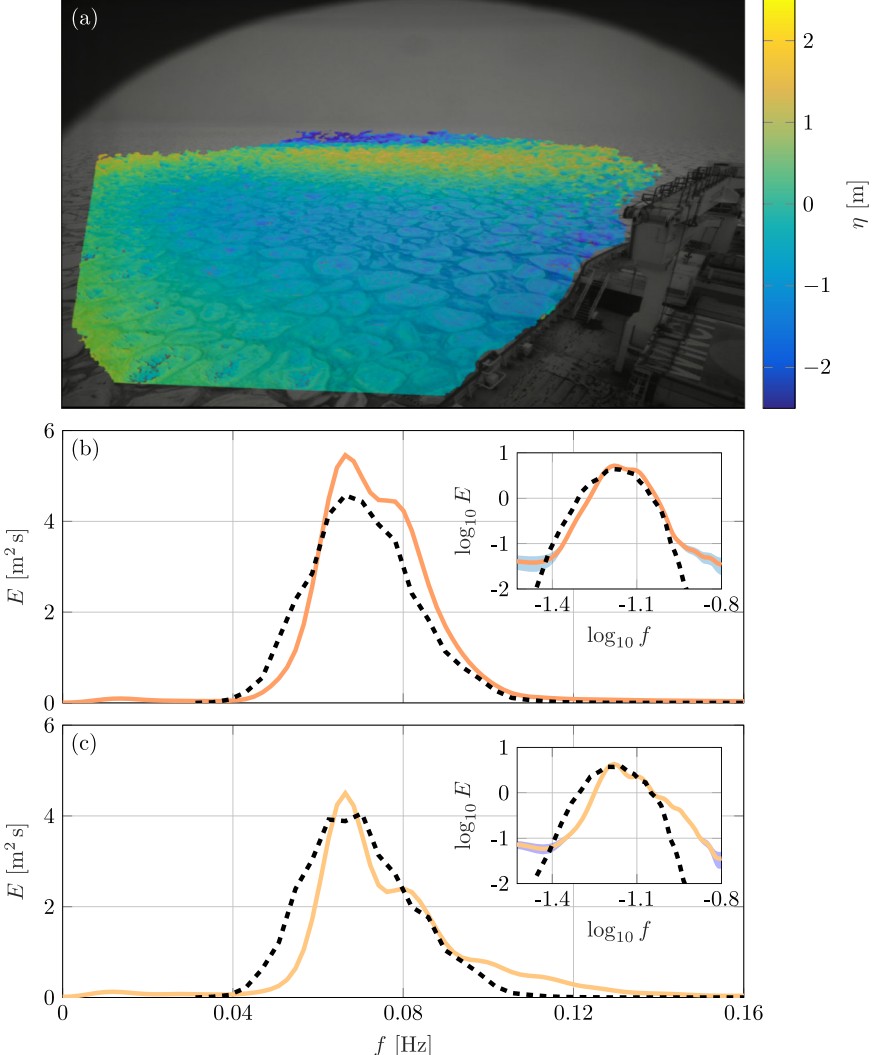

**Fig. 3 | Three-dimensional surface elevation retrieval and validation against buoys. a** Reconstructed surface elevation, $\eta$ (axes in pixels). **b, c** Frequency spectra were obtained from the surface elevations measured at the two deepest measurement locations ($d = 43$–44km; solid curves) and co-located buoy measurements (broken curves). Insets show the spectra in logarithmic scale and the shaded area denotes noise level associated to wave spectra derived from stereo images. **b, c** The horizontal axes denote frequency ($f$) and the vertical ones energy ($E$).

## Three-dimensional imaging of ocean surface and comparison with buoy data

The three-dimensional images (Fig. 3a) are used to extract ocean surface elevation timeseries, $\eta(t)$, at each measurement location, from which the one-dimensional frequency spectra, $E(f)$, and two-dimensional frequency–direction spectra, $E(f, \theta)$, are derived (see Methods). At the two deepest measurement locations ($d = 43$–44km), wave buoys (waves-in-ice observation systems[47] of the type used in the previous studies[11,18,20,21]), were deployed on ice floes[48] and the frequency spectra they provide are used to validate the analysis of the stereo-camera images (Fig. 3b, c). The overall shape of the corresponding spectra is consistent, noting that the buoys do not show the multiple peaks clearly due to a low resolution. Discrepancies occur for the lower ($f < 0.05$Hz) and upper tails ($f > 0.10$Hz), which show over and under estimation of energy, respectively, relative to the buoys. These modes carry a small amount of energy and make only minor contributions to integrated parameters, such as the significant wave height and mean period, for which values derived from the images and buoys differ by $\approx$5% and $\approx$2.5%, respectively.

The uncertainty associated with spectra derived from the stereo-camera images is primarily white noise (i.e., equal across any frequency band; see Methods). The noise level is negligible for the most energetic modes (0.04 Hz $< f <$ 0.16 Hz, or $-1.4 < \log_{10}(f) < -0.8$, corresponding to periods of $\approx$7–25s; see error bands in Fig. 3b, c insets) and produces an integrated error of $\approx$0.02m, which corresponds to -0.5% of the measured significant wave heights. The spectral density estimated from buoys is subjected to red noise (i.e., it decreases with increasing frequency), which also produces a negligible effect on the significant wave height[49].

## Surface elevation and wave spectra as a function of distance from the ice edge

Surface elevation timeseries around the largest individual waves recorded ($H_{max}$; maximum crest to trough distance) at the six measurement locations are shown in Fig. 4a. The significant wave height is also reported for each timeseries (Fig. 4a; horizontal dashed lines). The significant wave height decreases along the transect, from 6.6m close to the ice edge ($d = 5$km; top panel) to $H_S = 4.6$m at $d = 44$km (bottom panel), and, hence, wave energy attenuates over distance. Statistical analysis of the individual waves indicates consistency with Gaussian (linear) theory. For instance, the kurtosis (fourth order moment of the probability density function of the surface elevation and a measure of wave nonlinearity[50]) is close to three, similar to Gaussian sea states[50,51]. The maximum individual wave heights at each location, which are part

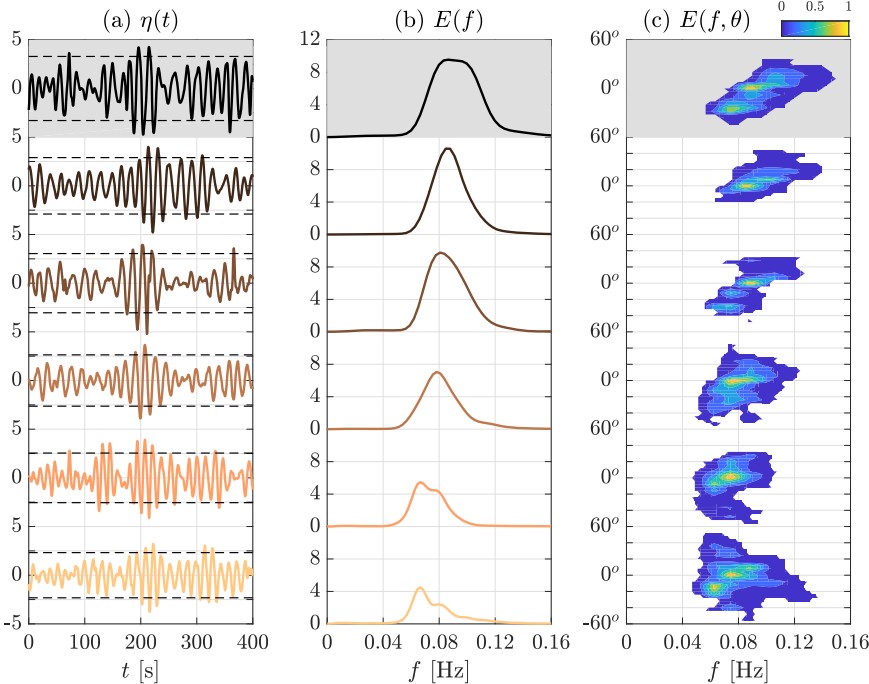

**Fig. 4 | Reconstructed surface elevation timeseries and wave spectra. a** Surface elevation timeseries at progressive distances from the ice edge (top–to–bottom; time on horizontal axes and surface elevation on vertical axes; colour-coding corresponds to locations shown in Fig. 3a) around the largest wave in each record. Dashed lines indicate significant wave heights, $H_S$. **b** Frequency and **c** frequency–direction wave spectra at progressive distances from the ice edge.

**b**, **c** The horizontal axis denotes frequency, in **b** the vertical axis wave energy and in **c** direction. Two-dimensional spectra are shown in Cartesian coordinates as the spectra cover only a narrow directional range, $-60° < \theta < 60°$, and normalised by the peak energy to highlight directional properties (colorbar shown next to the title). **a**–**c** Shaded backgrounds denote the measurement location in intermediate ice concentration.

of energetic wave groups, tend to diminish with distance into the MIZ. Nevertheless, large waves are recorded tens of kilometres into 100% ice concentration, e.g., $H_{max} \approx 8$m at $d = 43$km (second to bottom panel), which corresponds to $H_{max}/H_S \approx 1.6$, close to the maximum height expected in a linear sea state[43]. The steepness associated with the largest individual waves, $\varepsilon_{max} = \pi H_{max}/(2\lambda)$, where $\lambda$ is the wavelength, is a measure of the strength of the wave[42]. It decreases from $\varepsilon_{max} \approx 0.19$ at the ice edge ($d = 5$km) to $\approx 0.10$ at the deepest measurements locations ($d = 43$–44km). Despite the attenuation of the steepness over distance, these maximum values are expected to be large enough to have an impact on the ice cover, for example, by keeping the ice cover unconsolidated[46,52].

Attenuation of wave energy with distance into the MIZ is also evident in the one-dimensional spectra (Fig. 4b). As expected, higher frequencies (shorter periods) experience greater attenuation than lower frequencies (longer periods), causing narrowing of the spectral bandwidth (the breadth of the spectrum in frequency[42]), with the bandwidth at the two deepest measurement locations $\approx 80\%$ of the bandwidth close to the ice edge. The frequency spectrum is unimodal at the first four measurement locations ($d \le 36$km) but becomes bimodal at the deepest two measurement locations ($d = 43$–44km) where a low-frequency peak appears (peak period $\approx 15$s). The low-frequency peak indicates the swell initially north of the ice edge (Fig. 1c) catches up with the higher frequency wind sea generated close to the ice edge (peak period $\approx 12.7$s at $d = 36$km). The swell overlaps the wind sea system in frequency space, but the two systems are clearly separated in two-dimensional frequency–direction space (Fig. 4c) as they are travelling in different directions. The offset is $\approx 20°$, which is consistent with the difference in direction between wind sea and swell reported at the ice edge. For further analysis, the frequency–direction spectra are used to partition the total sea into swell and wind sea (see Methods). Note that the buoys do not return the directional spectrum[18] and, hence, their measurements cannot

be used to separate wind sea and swell systems, as they overlap in frequency space.

## Wave evolution

The wave age $c_p/u_{10}$[42], where $c_p$ is the phase velocity (ratio of wavelength to wave period) and $u_{10}$ is the wind speed, is computed at each measurement location using wind speeds from the onboard met-station (Fig. 5a). The values obtained indicate waves are young (growing in size and length under the action of wind) at the first four measurement locations ($c_p/u_{10} < 1.25$[53]). The waves switch sharply to old ($c_p/u_{10} > 1.25$) at the deepest two measurement locations. The sharp transition is partially due to the arrival of the swell system, but, as implied by the similar sharp transition in wave age for the wind sea (denoted total sea without swell), a sudden drop in wind speed from $\approx 25$ms$^{-1}$ to $\approx 17$ms$^{-1}$ is the primary cause.

Lengthening of the peak period with distance is evident in the frequency spectra (Fig. 4b). The peak period of the incident field also increases over the duration of the experiment, from 12.4s to 13.5s, and the measured peak period normalised by the corresponding ERA5 peak period at the ice edge to account for the changing conditions in the open ocean (based on the mean wave direction and group velocity of the mean period; see Methods), is relatively insensitive to distance until the swell system is detected at the deepest two measurement locations (Fig. 5b), indicating the peak period elongation due to preferential attenuation of shorter period components of the spectrum[21,54] is negligible. The normalised peak period sharply increases when the swell system appears at the deepest two measurement locations ($d = 43$–44km), but the increase is relatively small for the wind sea. The normalised peak period values for the wind sea are consistent with the MBK spectral attenuation model[15,21], where the ERA5 spectrum at the ice edge is used as the incident field for the model.

The frequency averaged directional spread ($\sigma_0$) is calculated from the two-dimensional spectra[20] and normalised using the ERA5 spectra

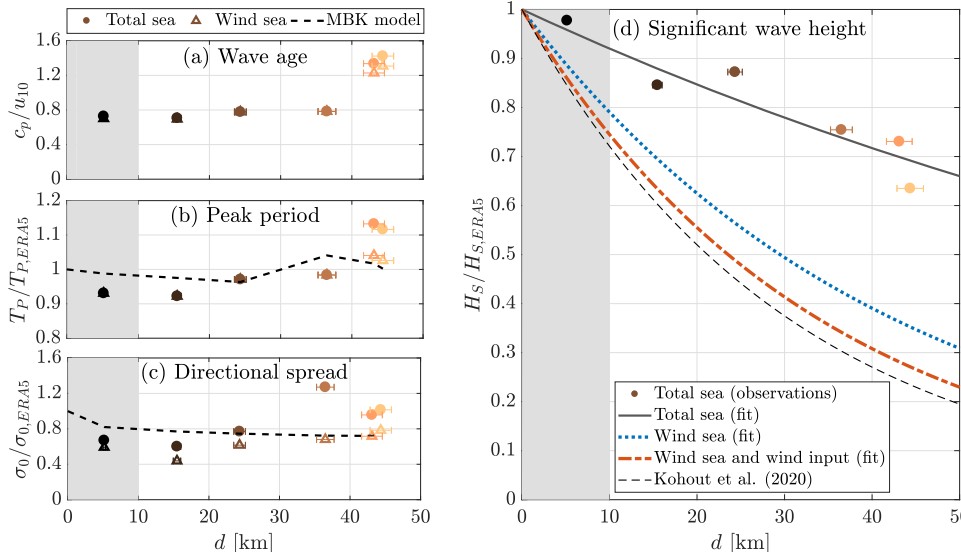

**Fig. 5 | Wave properties at progressive distances from the ice edge. a** Wave age; **b** peak period; **c** directional spreading; and **d** significant wave height (vertical axes) versus distance (horizontal axes). Dots are used to denote total sea, and triangles the wind sea (total sea without swell). The horizontal error bars denote uncertainties in distance from the ice edge due to variability in wind and wave directions. The shaded area denotes intermediate ice concentration. **b, c** The dashed line is derived by applying the MBK model[21] to corresponding ERA5 spectra at the ice edge. **d** The solid grey line denotes the best (exponential) fit for the total sea, the dashed line for the wind sea (total sea without swell) and the dash-dotted line the wind sea without wind input over ice. The thin dashed line is the benchmark attenuation derived for $i_c > 0.8$ and $T_P < 14$s[18].

at the ice edge (Fig. 5c). For the total sea, the normalised directional spread is relatively insensitive to distance over the first three measurement locations ($0.60 < \sigma_0/\sigma_{0,ERA5} < 0.77$) but increases sharply at the final three locations ($\sigma_0/\sigma_{0,ERA5} > 1$), with the maximum spread ($\sigma_0/\sigma_{0,ERA5} \approx 1.27$) at $d = 36$km. The wind sea shows little variation in normalised directional spread over all locations ($0.44 < \sigma_0/\sigma_{0,ERA5} < 0.88$). Thus, the peak in directional spread of the total sea at $d = 36$km is likely due to a combination of attenuation of the wind sea peak and emergence of the swell system. The directional spread of the total sea decreases at the deepest two locations as the swell dominates. The MBK attenuation model applied to the ERA5 two-dimensional incident spectra, with propagation distances of the spectral components dependent on their direction, predicts the normalised directional spread slightly decreases over distance from 50° ($\sigma_0/\sigma_{0,ERA5} = 0.82$) at 5km to 43° ($\sigma_0/\sigma_{0,ERA5} = 0.52$) at 43–44km, as the components travelling at oblique directions travel farther and, hence, experience greater attenuation leading to collimation[14]. This trend contrasts with the weak increasing trend in the observed directional spread of the wind sea by $\approx 1.22$% per kilometre.

The significant wave height of the total sea normalised by the ERA5 ice edge counterpart (to account for the non-stationary wave conditions at the ice edge, where the significant wave height grows from 6.7m to 7.4m during the measurements; see Methods) attenuates with distance (Fig. 5d). The best-fit exponential curve to the normalised measurements, $\exp(-\alpha d)$, gives the attenuation rate $\alpha = 8.3 \times 10^{-6}$m$^{-1}$. The wind sea attenuates at a greater rate, with the best-fit exponential curve giving the attenuation rate $\alpha = 23.5 \times 10^{-6}$m$^{-1}$. Subtracting the contribution of wind input to the wind sea (based on theory for open water and using winds recorded by the onboard met-station; see Methods) further increases the attenuation rate to $\alpha = 29.4 \times 10^{-6}$m$^{-1}$. For comparison, the benchmark empirical model[18] gives the rate $32.7 \times 10^{-6}$m$^{-1}$ (for ice concentrations greater than 80%, peak periods less than 14s and significant wave heights up to 6m). The uncertainty in the calculation of the distance from the ice edge (see error bars in Fig. 5), which is due to ambiguity introduced by the coexistence of multiple wave systems, does not affect the reported trends of wave parameters.

## Discussion

The benchmark significant wave height attenuation rate is greater than that derived for the total sea from the stereo-camera images by approximately a factor four. The benchmark rate is based primarily on measurements of low-energy sea states ($H_s < 1$m), where wind speeds were generally < 10ms$^{-1}$ (from ERA5 reanalysis; see Fig. 9 by Montiel et al.[23]) and the wave spectra were most likely unimodal (as indicated in Fig. 10c by Kohout et al.[18]). Therefore, we argue that the fairest comparison is with the wind sea without wind input (i.e., attenuation of the large waves generated by the cyclone in the open ocean), although noting the subtracted wind input represents an upper bound as it does not consider ice cover. With these modifications to the wave field, which rely on analysis of the frequency–direction spectrum, the attenuation rate is less than the benchmark by ≈10% only. The agreement is remarkable considering the potential differences in conditions that may affect attenuation rates, such as the extreme wave heights and strong winds associated with the cyclone, and the ice properties. Therefore, the results provide support for the benchmark attenuation rate and evidence it holds for considerably larger waves than previously recorded in similar ice conditions. However, the results show the benchmark attenuation rate only applies to single-component seas, and does not describe the evolution of the complex sea observed deep into the MIZ during the cyclone.

The results provide evidence that strong winds feed wave growth in 100% ice concentration for tens of kilometres over pancake/frazil ice cover. This contradicts the assumption made in most contemporary models that wind input scales according to the open water fraction[55,56], i.e., no wind input in 100% ice concentration. The assumption is already being debated, particularly for frazil, brash and/or pancake ice conditions[22], and theories for wind-to-wave momentum transfer through ice covers are being proposed[57].

The benchmark attenuation rate is greater than the attenuation rate of the wind sea without wind input possibly due to the properties of the ice cover. Based on observations during deployment of the buoys for the benchmark measurements, the ice cover consisted of unconsolidated pancake/frazil ice, similar to the stereo-camera measurements, and also consolidated, larger, thicker floes and continuous

ice[18], which is likely to cause stronger attenuation[58,59]. Building a larger database of stereo-camera images in the MIZ will drive improved understanding of how ice type influences the attenuation rate. Moreover, collecting stereo-camera images during conditions when the incoming wave field is steady will allow the spectral attenuation rate to be calculated and compared with benchmarks[21,23], thus giving more detailed understanding of the attenuation process.

In conclusion, measurements have been reported of extreme wave conditions in the winter Antarctic MIZ during an explosive polar cyclone, which were captured by a stereo-camera system on a moving vessel over a 44km transect and validated by co-located buoy measurements. The images empowered analysis of the frequency–direction wave spectrum evolution, and revealed the complex multi-component nature of the sea state in the MIZ where wind sea and swell co-exist and attenuate at different rates. Concomitant measurements of winds gave evidence of wind input through 100% ice cover. The success of stereo-imaging system shown in this study is likely to influence the design of future field campaigns in the MIZ. Moreover, it has the potential to be installed on vessels that routinely traverse the Antarctic MIZ and autonomously monitor the sea state, vastly increasing the data available as well as providing concomitant information on the ice cover. In turn, this will open new frontiers to advance current knowledge of MIZ dynamics and underpin the development of the next generation of climate models.

## Methods
### Image acquisition
The acquisition device consists of two GigE monochrome industrial CMOS cameras with a 2/3 inch sensor, placed side-by-side at a distance (i.e., baseline) of 4m. The stereo rig was installed on the monkey bridge of the icebreaker, ≈34m from the waterline and tilted 20° below the horizon. The cameras were equipped with 5mm lenses to provide a field of view of the ocean surface ≈90° around the port side of the ship (Fig. 3a). Additionally, an inertial measurement unit (IMU) was firmly attached close to the two cameras (between the cameras at the same height and ≈1m behind them) to capture the stereo-rig movement with respect to the sea surface during the acquisition. Images were recorded with a resolution of 2448 × 2048 pixels and a sampling rate of 2Hz during daylight on the 4 July 2017 (from 07:00 to 14:00 UTC).

### Floe size
An automatic algorithm, developed using the MatLab Image Processing Toolbox, was applied to extract sea-ice metrics from the recorded images. To avoid sampling the same floe twice, images every 10s were analysed. Images were ortho-rectified and projected on a horizontal plane. A camera-dependent calibration was applied to convert pixel to metres. The images were processed to eliminate the vessel from the field of view, adjust the image contrast and convert the grey scales into a binary map, which isolates the solid ice shapes from background water or frazil ice. A morphological image processing was applied to improve the fidelity of the shape of identified pancakes. Floe area ($S$) was calculated based on pixels and approximated by a disk, from which a characteristic diameter $D = \sqrt{4S/\pi}$ was defined. Further details on the algorithm and floe size distribution can be found in Alberello et al.[34].

### Estimation of the surface elevation
The Wave Acquisition Stereo System (WASS[60]) was used to estimate a set of data points in space (a dense 3D point cloud) representing the 3D ocean surface. WASS analyses left and right stereo images to find photometrically distinctive corresponding points (i.e., projections of the same 3D point in space) that can be triangulated to recover their original 3D position in space. The operation is performed for each pixel of the stereo frames to produce a temporal sequence of 3D point clouds composed of millions of samples each. The data is also automatically filtered to remove possible outliers and the mean sea-plane is estimated independently for each frame.

To be effective, the technique requires the geometrical configuration of the two cameras to be known a priori. WASS can estimate that property as part of the process, with the added advantage of correcting slight variations in the camera's reciprocal orientation due to vibrations. The motion of the vessel under the effect of waves, however, is more significant than vibrations induced, for example, by wind. It follows that points clouds from different pair of images lie in a different reference frame. Measurements of ship motion from the IMU are therefore used to align and geo-localise each cloud to a common horizontal plane defining the mean sea level. As the IMU does not estimate the absolute elevation accurately, an approach combining surface orientations estimated from the stereo data with the altitude computed by the IMU was developed to recover the camera motion throughout the sequence. An unscented Kalman filter is applied to model the six degrees of freedom position and orientation of the cameras (i.e., the system state) as a discrete-time random variable. At each frame, the system state is updated with both the absolute IMU data (yaw-pitch-roll) and the mean sea-plane distance vector estimated from the point clouds. Both are modelled as Gaussian distributions in which the measurement covariance is given by the sensor manufacturer specifications, in the case of the IMU, or the empirical stereo estimation error for the cameras[31]. With the estimated camera motion, each scattered point cloud is transformed on a common reference frame with the $x$–$y$ axes aligned with the mean sea-plane, and interpolated on a regular grid to reconstruct a timeseries of 3D surface elevations. The final dimension of the reconstructed surfaces is about 150m × 200m, which is ≈5 times larger than Smith & Thompson[37].

A systematic source of uncertainty is the resolution error[61], also referred to as quantisation noise, which depends on the object distance, the focal length and the cameras' resolution, mutual position and declination. An estimate of this error was calculated as the difference between a known synthetic surface and its "*back and forth*" transformation, which consists of projecting the known surface onto the camera coordinate system and re-projecting it onto the original coordinate system by using the specific geometry of the stereo-camera setup[61]. This difference provides the spatial distribution of the error, the amplitude of which is uniformly distributed across wavenumbers/frequencies (i.e., white noise).

### Estimation of the attenuation rate
The sea state conditions were non-stationary during the observation period, with wave height increasing ≈10% from 6.7m to 7.4m. Therefore, the attenuation rate $\alpha$ is estimated with respect to the ice edge and the dimensionless significant wave height according to the following equation:

$$\log\left(\frac{H_S}{H_{S,ERA5}}\right) = -\alpha d, \qquad (1)$$

where $d$ is the distance of each measurement location from the ice edge calculated along the mean wave direction in the open ocean as provided by the ERA5 reanalysis, and $H_S/H_{S,ERA5}$ is the significant wave height normalised by the incident (open ocean) counterpart from the ERA5 reanalysis. To estimate the incident wave conditions ($H_{S,ERA5}$), the delay between the time at which waves enter into the MIZ and the time at which waves are measured at a specific location was estimated through the wave group velocity. A standard least square fitting is applied to extrapolate an overall attenuation rate. Owing to the non-stationarity of the sea state conditions, each frequency components of the wave field is subjected to different forcing, besides ice-induced attenuation. Thereby, the estimate of a frequency-dependent attenuation rate[18] is impractical herein as it would be subjected to significant uncertainty.

Note that the coexistence of wind sea and swell systems generated an ambiguity for the selection of the relevant wave direction, which translates into uncertainties in the distance from the ice edge. By assuming an overall mean wave direction, the error associated to $d$ is ≈± 5% (this is reported as an horizontal error bar in Fig. 5).

## Frequency–direction spectrum

The directional wave spectrum can be computed from three measured quantities of the ocean surface (for example, a buoy uses either heave, pitch and roll or three accelerations) with a Fourier expansion method[62]. This approach produces accurate mean wave direction, but the directional spreading is typically too broad and with spurious bimodal properties. An improvement in the directional resolving power can be achieved by increasing the number of measured elements with a spatial array of sensors, which are cross-correlated using the maximum likelihood method or a wavelet directional method[62,63].

With the stereo images, the strategy is to apply an array approach to extract timeseries of the surface elevation and a wavelet directional method to approximate the directional spectrum, noting that the estimate of the spectrum from timeseries allows resolving components with periods > 11.5s and, hence, wavelength > 200m (i.e., longer than the field of view). A virtual array with geometry comprising of a triangle inscribed in a circle of radius ≈1m inside a pentagon inscribed in a circle of radius ≈3m and with an additional probe in the central position (a co-array configuration) was used to allow a sufficient number of non-redundant spatial lags between elements and asymmetry, which ensure an accurate estimation of the directional spreading function[62]. The wavelet directional method resolve modes around the spectral peak ($0.5 < f/f_p < 3$, where $f_p$ is the peak frequency) more accurately than the maximum likelihood method[63]. Thereby, it is the preferred approach to identify the complex multiple peak feature of the spectrum.

## Correction of Doppler shift and true wave spectrum

The sea surface elevation extracted from the images is in a frame of reference that moves with the forward speed and heading of the ship. The directional spectrum is therefore distorted due to a Doppler shift[64] and it is typically referred to as the encountered spectrum $E_S(f_S, \theta)$, where $f_S$ is the frequency detected by a moving object. To restore the original spectral shape, the encounter frequency is converted into the true frequency $f$ through the linear wave dispersion relation as follows[64]:

$$f_S = f + 4\pi^2 \frac{f^2}{g} U_S \cos\beta, \tag{2}$$

where $\beta$ is the angle between the ship heading and the (open water) mean wave direction from the ERA5 reanalysis. The true wave spectrum, $E(f, \theta)$, is computed by performing a change of variable:

$$E(f,\theta) = E_S(f_S,\theta)\frac{df_S}{df} = E_S(f_S,\theta)\left(1 + 8\pi^2 \frac{f}{g} U_S \cos\beta\right). \tag{3}$$

where $df_S/df$ is the Jacobian of the transformation.

## Partitioning of spectrum

The partitioning of the wave spectrum is performed using the path of steepest ascent technique[65], which is a specific implementation of the inverse catchment scheme introduced by Hasselmann et al.[66]. The spectral peak that satisfies the condition

$$1.2\frac{u_{10}}{c_p}\cos(\theta - \psi) > 1, \tag{4}$$

where $u_{10}$ is the wind speed, $c_p$ is the phase velocity, $\theta$ is the wave direction and $\psi$ is the wind direction, is assumed to be associated with the wind sea. All other systems are swell and are ranked based on their energy contents as primary, secondary and tertiary swell. Only primary swell was considered.

## ERA5 incident sea state

The incident sea state at the ice edge is obtained from ERA5 reanalysis, which provides hourly wave data at a resolution of ≈40km. The model performs well in the Southern Ocean but can miss the swell arrival time[67] and the shape of the spectrum is limited by directional resolution (15°), which makes it difficult to discriminate wind seas and swell that propagate in approximately the same direction.

The ERA5 wave model uses a simplistic wave attenuation schemes in sea ice, which acts only at the outskirts of the MIZ (up to 15% ice concentration), and assumes waves are completely dissipated for ice concentration > 15%. To overcome uncertainties related to the model assumptions, we adopt, as a reference, the wave data in open ocean (0% ice concentration), in the proximity of the ice edge along the mean wave direction (the distances from the ice edge specified in Fig. 2a are the one projected in the direction of the mean wave direction). We also account for the time delay due to wave energy advection (based on the wave group velocity from the incident ERA5 mean period) to associate sea states in the MIZ with the open ocean counterpart.

## Comparison against ERA5 reanalysis

To provides evidence of consistency between the ERA5 reanalysis and the observations presented herein, and thus to allow using the former to produce incident sea states, a comparison against open ocean conditions recorded on July 2nd (between 12:00 and 14:00 UTC at 52° South and 26° East) and July 3rd (between 12:00 and 13:00 UTC at 56° South and 28° East) is shown in Table 1. For this comparison, the ERA5 reanalysis data refers to the value at the closest grid point, while observations are an average over the time period. Overall the agreement is good, with difference in wave height < 0.2m (< 4%), wave period < 0.2s (< 2%) and directional spread <0.3°. A more thorough evidence of the accuracy of the ERA5 reanalysis in the Southern Ocean through comparison against a large data set acquired during the Antarctic Circumnavigation Expedition is detailed by Derkani[68].

## MBK attenuation model

Based on field measurements in the Antarctic MIZ, MBK (Meylan et al.[21]) proposed an (exponential) attenuation rate for each frequency component in the spectrum as

$$\beta(f) = af^2 + bf^4 \quad \text{where} \quad a = 2.12 \times 10^{-3} \quad \text{and} \quad b = 4.59 \times 10^{-2}. \tag{5}$$

The MBK attenuation rate is used to propagate the incident directional wave spectrum into sea ice up to the measurements locations using the expression

$$E(f,\theta) = E_{ERA5}(f,\theta)\exp(-i_c\beta(f)d(\theta)). \tag{6}$$

**Table 1 | Integrated spectral wave parameters from our measurements and ERA5**

| - | $H_{S,WASS}$ [m] | $H_{S,ERA5}$ [m] | $T_{P,WASS}$ [s] | $T_{P,ERA5}$ [s] | $\sigma_{O,WASS}$ [°] | $\sigma_{O,ERA5}$ [°] |
|---|---|---|---|---|---|---|
| 2nd July 2017 | 4.93 | 4.94 | 10.18 | 10.37 | 35.6 | 35.9 |
| 3rd July 2017 | 4.16 | 4.35 | 8.01 | 7.84 | 36.3 | 36.2 |

The attenuation rate depends on the ice concentration ($i_c$), the frequency of the wave component and the distance from the ice edge on the direction of each component.

## Wind input

In the open ocean, wind transfers momentum to the sea surface, forcing ocean waves to grow over distance (fetch). Wave growth is estimated with empirical models, which are expressed as[42]

$$\tilde{H}_S = a_1 \tilde{F}^{b_1} \tag{7}$$

where $\tilde{F} = gF/u_{10}^2$ and $\tilde{H}_S = gH_S/u_{10}^2$ are the dimensionless fetch and wave height, respectively, and $a_1 = 2.88 \times 10^{-3}$ and $b_1 = 0.45$[69]. Eqn. (7) is used to estimate the fetch at the ice edge ($F_{ERA5}$) from the known ERA5 wave height ($H_{S,ERA5}$). The distance from the edge ($d$) is added to the fetch

$$F = F_{ERA5} + d. \tag{8}$$

Eqn. (7) is then applied again to give an updated estimate of the wave height ($H_{S,wind}$) over the entire distance ($F$). The difference between the updated wave height and the one at the edge provides an estimate of the wave height attributed to wind input, i.e.

$$\Delta H_S = H_{S,wind} - H_{S,ERA5}. \tag{9}$$

## Data availability

Data sets for this research (reconstructed surface elevations) are available through the Australian Antarctic Data Centre (AADC): Alberello et al. (2021) Wave acquisition stereo-camera system measurements (WASS) from a voyage of the S.A. Agulhas II, July 2017, Ver. 1, Australian Antarctic Data Centre–https://doi.org/10.26179/q9bd-5f74.

## Code availability

MATLAB was used for the analysis. Data processing is described within the manuscript and code is available from the first author upon request.

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

## Acknowledgements

The expedition was funded by the South African National Antarctic Programme through the National Research Foundation. This work was motivated by the Antarctic Circumnavigation Expedition (ACE) and partially funded by the ACE Foundation and Ferring Pharmaceuticals. A.A., L.B. and A.T. were supported by the Australian Antarctic Science Programme (project 4434). A.A. acknowledges support from the Japanese Society for the Promotion of Science (PE19055). L.G.B. is supported by the Australian Research Council (FT190100404). L.G.B. and A.T. are supported by the Australian Research Council (DP200102828). M.O. was supported by the Simons Collaboration on Wave Turbulence, Award No. 617006, and from the "Departments of Excellence 2018-2022" Grant awarded by the Italian Ministry of Education, University and Research (MIUR, L.232/2016). M.O. acknowledges the EU H2020 FET Open BOHEME, Grant No. 863179. M.V. and K.M. were supported by the NRF SANAP contract UID118745. C.E. was supported under NYUAD Center for

Global Sea Level Change project G1204. We are indebted to Captain Knowledge Bengu and the crew of the S.A. Agulhas II for their invaluable contribution to data collection. ERA5 reanalysis was obtained using Copernicus Climate Change Service Information. M.O. acknowledges B. GiuliNico for interesting discussions. A.A., A.T. and M.O. thank L. Fascette for technical support during the cruise.

## Author contributions

Conceptualisation: A.A., L.G.B., M.O., A.T. Methodology: A.A., L.G.B., M.O., M.V., K.M.H., C.E., B.N.N., A.B., F.B., F.N., R.P., H.C., A.T. Investigation: A.A., L.G.B., M.O., M.V., A.T. Visualisation: A.A., L.G.B., I.T., A.T. Supervision: A.A., L.G.B., A.T. Writing-original draft: A.A., L.G.B., M.O., M.V., A.B., F.B., A.T. Writing-review & editing: A.A., L.G.B., M.O., M.V., K.M.H., C.E., B.N.N., A.B., F.B., F.N., R.P., H.C., I.T., A.T.

## Competing interests

The authors declare no competing interests.
