## [Peer Review File · Nature Communications]

Three-dimensional imaging of waves and floe sizes in the marginal ice zone during an explosive cycloneREVIEWER COMMENTS

Reviewer #1 (Remarks to the Author):

“Three-dimensional imaging of waves in the marginal ice zone during an explosive cyclone” by Alberto Alberello et al. presents novel measurements of interaction of waves with sea ice in the Antarctic marginal ice zone. This is a very active area of research, and it is nice to see wave-ice work presented at this level. Although stereo-video measurements of ocean waves are becoming more routine, in this instance, it’s a unique deployment where they measure directional wave properties in high ice concentration conditions. The writing is error-free, the figures are polished and high quality. From this reviewer’s point of view, this paper represents a technical achievement that has potential to be implemented in a routine way. On the other hand, it is a little short on novel ideas, analysis, and findings. Nevertheless, I would consider this manuscript for publication if the measurement technology was more fully vetted and several key issues are addressed.

If you have the stereo-video, why isn't it being used to say more about the ice properties? E.g. the floe size distribution? If there was a story to be found there, I think it would be a very powerful application.

(Paragraph 1) Add a statement about how this technology addresses this need.

(Paragraph 2) These blanket statements do not carry enough nuance. The only statement I fully agree with is your point around momentum transfer. I would like to see you work harder to convey information about the state-of-the-art and where the issues lie. E.g. One might say that although there are a number of models that predict wave attenuation (see any of Vernon Squire’s review articles), many of which are included in operational spectral wave model WAVEWATCHIII (see the most recent manual v6.07), we still don’t have enough information about the ice (concentration, thickness, rheology) to apply the more detailed and complex treatments of wave-ice interaction. So we revert to simple empirical solutions.

(Paragraph 3) References for Arctic and Antarctic are swapped. I am not usually one to insist on certain citations, but I think Thomson et al. (2018) deserves mention here. These statements about buoys are at best confusing and at worst misleading. Maybe you are referring specifically to the buoys used in this study, but there are all kinds of buoys capable of all kinds of measurements. Some of these buoys can be used in any ice concentration. Doble and Bidot (2013) report on buoys that were frozen into the ice, then eventually freed (obviously not destroyed). Some buoys use GPS some IMUs. You use the word “reconstruction” twice, neither is fitting. Buoys don’t reconstruct waves, they measure some property of the sea surface (either acceleration, speed, or displacement). The sea surface elevation is simply and directly related to a measured property. “They suffer from poor sensitivity to small accelerations, which severely hampers reconstruction of the directional wave spectrum”. Assuming you are referring to accelerometer buoys, then yes, there are ranges in a spectrum (typically outside the energetic wave range) where the signal to noise ratio can be low, which on the signal. Again, the directional wave spectrum is not “reconstructed”, a buoy measures properties of the sea surface that are directly related to the first 5 Fourier coefficients of the full directional spectrum. This is generic to single-point-triplet systems and has nothing to do with signal to noise ratio of accelerometer buoys. Since the first 5 coefficients is a low order estimation of the full directional spectrum, this could be a place to highlight the advantages of stereo-video. I assume the stereo-video has a higher order (more coefficients) but you don’t directly address this. Since you designed a synthetic array, you could quote the resolution and how this may be better than a typical buoy. Overall, this reads like the authors are not very familiar with how buoys operate, and are working hard little too hard to emphasize their perceived disadvantages in order to prop up stereo-video technology. It would be better to more plainly discuss pros and cons of each, your measurement technique is very interesting

and stands on its own. There is no need to implicate as a bad way to measure waves, especially since they have been the workhorse for making discoveries in wave-ice interaction. Indeed, you later use buoys to validate your own measurements. (If you insist, then at least cite something like Thomson et al. (2021) that looks at how buoy noisy measurements can be misinterpreted.)

(Paragraph 4 - last statement) Move the end to the front: "In contrast to SAR, stereo-video..."

(Results - Last paragraph and Figure 2) Plotting spectra on top of one another is a very weak form of validation, especially considering they don't seem to match up very well. I would like to see log-log plots. Is there any validation outside of ice you might point to?

(Surface elevation and wave spectra - first paragraph) Hmax is a function of time (number of waves measured), unless you control for the number of waves measured, Hmax is not an apples to apples metric to compare with distance into MIZ. "These are the largest waves recorded in 100% ice concentration". Neat, but why is this superlative important? What does it add to your story? (They don't report Hmax nor to and I think the ice concentrations were uncertain, but Doble and Bidlot (2013) would have almost certainly measured larger waves.)

(Surface elevation and wave spectra - first paragraph) This is such a prominent feature of wave interaction reported in the literature, in the second line you might add "As expected, higher..."

(Wave evolution paragraph 3) The assumption for all of this is that the ERA5 estimates are comparable to measurements for the metric considered. You should probably quote, if only in the Methods section, some typical differences between ERA5 and measurements. Hm0 and Tp are probably good, what worries me more is directional spread. I'm not so sure there is good correspondence between model and measured directional spread in general.

(Wave evolution paragraph 3) "and the model-data agreement for $d = 36-44$ km is misleading." Are you saying you got the right answer for the wrong reasons? Can you elaborate?

(Discussion paragraph 1) It took me several readings to follow what you are arguing here, but I think I finally understand and I think it makes sense. So besides the specific questions and feedback below, I suggest working on writing here so that the points you want to make follow a logical progression and make the connections to the ideas you want to support. First, you are going to have to justify why that study should be considered a benchmark. "where multiple-component seas and large wind input are unlikely." This is not good enough, if you can look up ERA5 for your study, surely you can go back to the ERA5 hindcast and show it for that one. Attenuation of Hm0 is the simplest possible approach, and I don't understand the utility of this approach because of the sensitivities to all the stuff you mention 1) the incident wave spectra (both frequency and directional-frequency), 2) the ice properties, 3) and potential wind input. I sort of get the point: that the measurements don't match up to previous measurements if you just broadly apply it, but they start to converge if you account for wind input and take out the swell (which maybe you can only do because of the unique stereo-video measurement?) Which is sort of neat because it shows off the utility of the stereo-video, but also sort of shows why the Hm0 approach is so problematic to begin with. You can hand wave away any disagreement, making agreement seem more coincidental.

Can you see swell in the buoy measurements?

Could you possibly give attenuation as a function of frequency, like plot like figure 17 of Kohout et al. (2020)? Are there too few measurements?.

(Methods - Surface elevation reconstruction). "...we remove the effects due to the vessel's forward speed..." This is not a straightforward problem, and the shape of the spectrum tends to be sensitive to how this is done. I need a detailed explanation here. How exactly was this done? What equations were used? What in that reference did you use? Did you consider data while the ship was accelerating or turning?

References:

Doble, M.J. and Bidlot, J.R., 2013. Wave buoy measurements at the Antarctic sea ice edge compared with an enhanced ECMWF WAM: Progress towards global waves-in-ice modelling. *Ocean Modelling*, 70, pp.166-173.

Thomson, Jim, Stephen Ackley, Fanny Girard-Ardhuin, Fabrice Ardhuin, Alex Babanin, Guillaume Boutin, John Brozena et al. "Overview of the arctic sea state and boundary layer physics program." *Journal of Geophysical Research: Oceans* 123, no. 12 (2018): 8674-8687.

Thomson, J., Hošeková, L., Meylan, M.H., Kohout, A.L. and Kumar, N., 2021. Spurious rollover of wave attenuation rates in sea ice caused by noise in field measurements. *Journal of Geophysical Research: Oceans*, 126(3), p.e2020JC016606.

TWIDG, W., 2019. User manual and system documentation of Wavewatch III version 6.07. tech note 333. Tech. rep., NOAA/NWS/NCEP/MMAB.

Squire, V.A., Dugan, J.P., Wadhams, P., Rottier, P.J. and Liu, A.K., 1995. Of ocean waves and sea ice. *Annual Review of Fluid Mechanics*, 27(1), pp.115-168.

Squire, V.A., 2007. Of ocean waves and sea-ice revisited. *Cold Regions Science and Technology*, 49(2), pp.110-133.

Squire, V.A., 2020. Ocean wave interactions with sea ice: a reappraisal. *Annual Review of Fluid Mechanics*, 52, pp.37-60.

Reviewer #2 (Remarks to the Author):

This paper presents a novel dataset of waves propagating in the marginal ice zone of the Southern Ocean during an explosive cyclone. I see two main points that this paper makes. The first one is the demonstration that ship-based photogrammetry, and stereoscopy in particular, is a promising method for characterizing waves propagating in an ice cover. Indeed the method successfully measures very large waves quite accurately, with respect to independent measurements obtained with wave buoys, and it is described such that it can be implemented elsewhere. It is worth noting that the WASS is an open source software that has been used in many other applications. The second point is that attenuation rates comply with what's been measured and empirically modeled in previous studies, and that wind can be responsible for wave growth in 100% ice concentration. The latter is based on wave attenuation coefficients that were measured with wave buoys during another experiment in the Antarctic MIZ, and on a number of assumptions about how wave attenuation is calculated in this paper and how wind-wave generation occur. Those two results are relatively well conveyed and deserve to be published.

I see however some weaknesses in the paper that can be improved before publication. The comment can be summarized like this: wave-ice interactions are so complex, non-linear and variable that concluding about what's going on in a particular situation necessitates taking into account the possible errors that can influence the outcome. First, there is very little information about sea ice along the transects, even though an imaging system was used to characterize waves-in-ice. The scientific literature arguing that thickness, floe size and other properties like ice rigidity affect wave attenuation is

abundant, either theoretically and observationally. The discussion about wave attenuation suggests to the reader that the attenuation solely depends on ice concentration, and wave frequency. Stopa et al. (2018) (<https://doi.org/10.1073/pnas.1802011115>), using a very large data set obtained from satellite radar imagery, provide evidence that the attenuation rate in ice is highly variable and cover many orders of magnitude. Given that cameras were used, the results should mention how the ice properties evolved along the transect, and the discussion should be adjusted at least to acknowledge the role of ice properties in setting the attenuation rate.

A second weakness lies in the calculation of the frequency wave spectra. Figures 2 and 3 do not show the noise level from the buoy, that is typically red, and from the stereo imaging. Typically also these figures are shown on a log scale. Thomson et al. (2021) (<https://doi.org/10.1029/2020JC016606>) describe a procedure that avoids contaminating attenuation results with noise. This should be discussed, especially when comparing data obtained from different sources (stereo and buoys). The footprint of the camera system is approximately 200m. What happens for wave component having a wavelength larger than this? Measuring the amplitude of such of wave would require a good knowledge of the absolute height, which is not straightforward using an IMU. Could the authors comment on this? Does that explain why the energy for long waves is less when measure with the camera than with the buoy (Fig. 2)?

The directionality of the wave spectra is another source of uncertainty that should be taken into account. In the current discussion, the energy is partitioned following the local wind, if I understand the algorithm correctly. In order to compute an attenuation coefficient between two stations, waves reaching the innermost station must have passed by the outermost station. Can the adequate partitioning target the direction connecting the two stations rather than the direction of the local wind seas, which according to Fig. 1 seems to be turning quite significantly?

Since the conclusion about wind-generation depends on Fig. 4, which do not show uncertainties and which is still quite speculative, I strongly recommend that the various sources of uncertainties be better quantified, illustrated and discussed. Otherwise, the paper is very well written and I fully recognize the importance of this work, as well as the challenges that analyzing these data poses.

Response to Referee #1

We thank the Referee for the insightful comments, which have helped us improve the manuscript. Our comment-by-comment response follows.

“Three-dimensional imaging of waves in the marginal ice zone during an explosive cyclone” by Alberto Alberello et al. presents novel measurements of interaction of waves with sea ice in the Antarctic marginal ice zone. This is a very active area of research, and it is nice to see wave-ice work presented at this level. Although stereo-video measurements of ocean waves are becoming more routine, in this instance, it’s a unique deployment where they measure directional wave properties in high ice concentration conditions. The writing is error-free, the figures are polished and high quality. From this reviewer’s point of view, this paper represents a technical achievement that has potential to be implemented in a routine way. On the other hand, it is a little short on novel ideas, analysis, and findings. Nevertheless, I would consider this manuscript for publication if the measurement technology was more fully vetted and several key issues are addressed.

If you have the stereo-video, why isn’t it being used to say more about the ice properties? E.g. the floe size distribution? If there was a story to be found there, I think it would be a very powerful application.

We agree with the Referee that inclusion of sea ice properties would enhance the quality of the presentation and, hence, we have added this analysis on floe sizes in a new figure (Figure 2 in the revised manuscript) and accompanying text (final paragraph of the Experimental Conditions section). Image processing techniques to retrieve sea ice properties during the same cruise were reported by Alberello et al. [2019] and only a short summary of it is added to the Methods (see Floe Size section).

(Paragraph 1) Add a statement about how this technology addresses this need.

(Paragraph 2) These blanket statements do not carry enough nuance. The only statement I fully agree with is your point around momentum transfer. I would like to see you work harder to convey information about the state-of-the-art and where the issues lie. E.g. One might say that although there are a number of models that predict wave attenuation (see any of Vernon Squire’s review articles), many of which are included in operational spectral wave model WAVEWATCHIII (see the most recent manual v6.07), we still don’t have enough information about the ice (concentration, thickness, rheology) to apply the more detailed and complex treatments of wave-ice interaction. So we revert to simple empirical solutions.

(Paragraph 3) References for Arctic and Antarctic are swapped. I am not usually one to insist on certain citations, but I think Thomson et al. (2018) deserves mention here. These statements about buoys are at best confusing and at worst misleading. Maybe you are referring specifically to the buoys used in this study, but there are all kinds of buoys capable of all kinds of measurements. Some of these buoys can be used in any ice concentration. Doble and Bidot (2013) report on buoys that were frozen into the ice, then eventually freed (obviously not destroyed). Some buoys use GPS some IMUs. You use the word “reconstruction” twice, neither is fitting. Buoys don’t reconstruct waves, they measure some property of the sea surface (either acceleration, speed, or displacement). The sea surface elevation is simply and directly related to a measured property. “They suffer from poor sensitivity to small accelerations, which severely hampers reconstruction of the directional wave spectrum”. Assuming you are referring to accelerometer buoys, then yes, there are ranges in a spectrum (typically outside the energetic wave range) where the signal to noise ratio can be low, which on the signal.

In response to the comments from both Referees, particularly those on the key contributions of the work, we have re-written the opening three paragraphs of the Introduction and made corresponding modifications to the remaining paragraphs. In particular, we have:

- Added a statement at the end of the first paragraph on how the stereo-camera technology addresses the need for autonomous monitoring of waves and ice in the MIZ.

- Focussed on the rapid advances made over the past decade in understanding wave attenuation in the MIZ, which are largely due to the improved observational capabilities given by buoy arrays.
- Included paragraphs on state-of-the-art for in situ observations (paragraph 2) and attenuation models (paragraph 3).

Again, the directional wave spectrum is not “reconstructed”, a buoy measures properties of the sea surface that are directly related to the first 5 Fourier coefficients of the full directional spectrum. This is generic to single-point-triplet systems and has nothing to do with signal to noise ratio of accelerometer buoys. Since the first 5 coefficients is a low order estimation of the full directional spectrum, this could be a place to highlight the advantages of stereo-video. I assume the stereo-video has a higher order (more coefficients) but you don’t directly address this. Since you designed a synthetic array, you could quote the resolution and how this may be better than a typical buoy. Overall, this reads like the authors are not very familiar with how buoys operate, and are working hard little too hard to emphasise their perceived disadvantages in order to prop up stereo-video technology. It would be better to more plainly discuss pros and cons of each, your measurement technique is very interesting and stands on its own. There is no need to implicate as a bad way to measure waves, especially since they have been the workhorse for making discoveries in wave-ice interaction. Indeed, you later use buoys to validate your own measurements. (If you insist, then at least cite something like Thomson et al. (2021) that looks at how buoy noisy measurements can be misinterpreted.)

Young [1994] provides a comprehensive study on the differences between various methods to estimate the directional wave spectrum, including the three components of motion and arrays. In the revised manuscript, we have added a brief discussion detailing the differences between the buoy approach (i.e. the three component of motion) and the image-analysis approach (i.e. the virtual array) in the Frequency–Direction Spectrum section of Methods, referring to the findings of Young [1994]. In the same section, we have provided details on the specific array geometry and commented on the resolution. The use of the term “reconstructed” has been removed.

(Paragraph 4 - last statement) Move the end to the front: “In contrast to SAR, stereo-video...”

We have made the suggested change.

(Results - Last paragraph and Figure 2) Plotting spectra on top of one another is a very weak form of validation, especially considering they don’t seem to match up very well. I would like to see log-log plots. Is there any validation outside of ice you might point to?

We have included log–log plots of wave spectra as insets to the linear plots (see Figure 3 of the revised manuscript). Error bands have also been included to show uncertainties. The new figure highlights that the energy distribution around the peak is consistent with the buoy, although some discrepancies are evident in the spectral tails. Energy associated to uncertain modes is small and discrepancies in the estimate of significant wave height are also small (less than 5% relative to buoys). We have updated the discussion of the figure in the Three-dimensional Imaging of Ocean Surface and Comparison with Buoy Data section.

Validation of the image processing technique (WASS) in open ocean has been carried out using measurements from installation on fixed platforms. Details of the validation are reported in, for example, Guimarães et al. [2020]. A comment on open water validation and reference to Guimarães et al. [2020] have been included in the Estimation of the Surface Elevation section of Methods.

(Surface elevation and wave spectra - first paragraph) Hmax is a function of time (number of waves measured), unless you control for the number of waves measured, Hmax is not an apples to apples metric to compare with distance into MIZ. “These are the largest waves recorded in 100% ice concentration”. Neat, but why is this superlative important? What does it add to your story? (They don’t

report H_{max} nor to and I think the ice concentrations were uncertain, but Doble and Bidlot (2013) would have almost certainly measured larger waves.)

The analysis of the individual waves adds to the story of the manuscript. In particular, we discuss statistical properties of the surface elevation, such as the kurtosis, to demonstrate that the wave field is a Gaussian random process. Knowledge of the wave statistics (rather than simply bulk values) is useful to understand impacts of the wave field on the ice cover, such as maintaining an unconsolidated ice cover [e.g. Womack et al., 2022]. The maximum individual wave height (H_{max}) is a statistical metric that gives a benchmark for future measurements at tens of kilometres into the MIZ in 100% ice concentration. In the revised manuscript (Surface Elevation and Wave Spectra as a Function of Distance from the Ice Edge section), we have also used the H_{max} -values to give further evidence of Gaussian statistics, by stating that $H_{max}/H_S \approx 1.6$, which is close to the maximum individual wave height $\approx 1.7 H_S$ expected for Gaussian random process. Moreover, we give the corresponding steepness values, which are typically used when assessing impacts of waves on ice cover. We have removed the superlative for the large waves, which are not central to the story. Lastly, we note that 20 minute time-series are consistent with length requirements for wave statistics given by the World Meteorological Organisation [World Meteorological Organization, 1998].

(Surface elevation and wave spectra - first paragraph) This is such a prominent feature of wave interaction reported in the literature, in the second line you might add "As expected, higher..."

We have made the suggested change (assuming the Referee meant the second paragraph).

(Wave evolution paragraph 3) The assumption for all of this is that the ERA5 estimates are comparable to measurements for the metric considered. You should probably quote, if only in the Methods section, some typical differences between ERA5 and measurements. H_{m0} and T_p are probably good, what worries me more is directional spread. I'm not so sure there is good correspondence between model and measured directional spread in general.

A comprehensive comparison of ERA5 reanalysis against in-situ measurements from a marine radar in the Southern Ocean is reported by Derkani [2021]. She shows good agreement across a variety of parameters, including significant wave height, mean period, mean wave direction and mean directional spread. A brief note about the results given by Derkani [2021] has been added to the Methods (see Comparison Against ERA5 Reanalysis section). Further, we included a table in the Methods (Comparison against ERA5 Reanalysis section), showing a direct comparison between our observations of significant wave height, mean period and directional spread and ERA5 reanalysis for the 2nd and 3rd of July (i.e. open ocean conditions two and one days before reaching the marginal ice zone, respectively). The agreement is excellent with differences in directional spread lower than 0.5 degree.

(Wave evolution paragraph 3) "and the model-data agreement for $d = 36-44$ km is misleading." Are you saying you got the right answer for the wrong reasons? Can you elaborate?

We were implying that the MBK attenuation model predicting the correct directional spread (of the wind sea) at the deepest locations is a coincidence. The MBK model predictions are included in Fig. 5c (the directional spread results, formerly Fig. 4c) mainly to highlight that the data do not back the model prediction of narrowing of the directional field due to collimation. We have revised the final sentence of the paragraph to clarify this.

(Discussion paragraph 1) It took me several readings to follow what you are arguing here, but I think I finally understand and I think it makes sense. So besides the specific questions and feedback below, I suggest working on writing here so that the points you want to make follow a logical progression and make the connections to the ideas you want to support.

We have rewritten the first three paragraphs of the Discussion to give a more logical flow to our arguments, based on the key points identified by the Referee.

First, you are going to have to justify why that study should be considered a benchmark.

We have justified the empirical model of Kohout et al. [2020] as a benchmark in the final paragraph of the Introduction on the basis that the model is derived from a very large dataset, which was collected during a similar season to ours. We have also taken this opportunity to state that the benchmark is used to argue existence of momentum transfer from winds.

“where multiple-component seas and large wind input are unlikely.” This is not good enough, if you can look up ERA5 for your study, surely you can go back to the ERA5 hindcast and show it for that one.

The ERA5 hindcast would not show the wave field in the MIZ. However, our statement about the likely conditions during the benchmark measurements is backed by the unimodality of the example wave spectra for low-energy seas reported by Kohout et al. [2020] (their Fig. 10) and the recently published study using the same dataset by Montiel et al. [2022] (their Fig. 2), and that Montiel et al. [2022] (their Fig. 9) show that the majority of measurements are for wind speeds $< 10 \text{ m s}^{-1}$ (from ERA5 reanalysis).

We have included this information in the Discussion.

Attenuation of H_m0 is the simplest possible approach, and I don't understand the utility of this approach because of the sensitivities to all the stuff you mention 1) the incident wave spectra (both frequency and directional-frequency), 2) the ice properties, 3) and potential wind input. I sort of get the point: that the measurements don't match up to previous measurements if you just broadly apply it, but they start to converge if you account for wind input and take out the swell (which maybe you can only do because of the unique stereo-video measurement?) Which is sort of neat because it shows off the utility of the stereo-video, but also sort of shows why the H_m0 approach is so problematic to begin with. You can hand wave away any disagreement, making agreement seem more coincidental.

Studying attenuation in terms of the significant wave height is logical for transient incident wave conditions created by the cyclone (see response to related comment on spectral attenuation below). We have justified this form of analysis in the text accompanying Fig. 5d, and adapted the discussion of the significant wave height versus spectral attenuation analysis in the Introduction (end of paragraph 2) and Discussion (end of paragraph 3).

Can you see swell in the buoy measurements?

Frequency and directional resolutions for the buoy measurements are not sufficient to isolate the different wave systems clearly. This is a known problem for this class of buoy, e.g. Kohout et al. [2020] state “[buoys] cannot return wave directional spectra”. We added a comment at the end of the Surface Elevation and Wave Spectra as a Function of the Distance from the Ice Edge section.

Could you possibly give attenuation as a function of frequency, like plot like figure 17 of Kohout et al. (2020)? Are there too few measurements?

Analysis of the attenuation as a function of frequency requires stationarity of all spectral components. However, the environmental conditions during the experiment are not stationary, due to wind input, coexistence of multiple systems and, above all, increasing incident wave intensity, making a frequency-dependent analysis of the attenuation rate uncertain. We have added a comment to explain the non-stationarity of the wave field in the revised version of the manuscript (see Experimental Conditions section).

(Methods - Surface elevation reconstruction). “...we remove the effects due to the vessel's forward speed...” This is not a straightforward problem, and the shape of the spectrum tends to be sensitive to how this is done. I need a detailed explanation here. How exactly was this done? What equations were used? What in that reference did you use? Did you consider data while the ship was accelerating or turning?

This is standard procedure in marine hydrodynamics and it is explained in e.g. Faltinsen [1993]. The method requires (i) the correction of frequencies to remove the Doppler shift, and (ii) a change of variable for the spectral density function. The approach also requires the assumption that, during the 20 mins over which the analysis is conducted, the ship forward speed and heading are constant. We have added details in the Correction of Doppler Shift and True Wave Spectrum section of Methods.

References

- A. Alberello, M. Onorato, L. Bennetts, M. Vichi, C. Eayrs, K. MacHutchon, and A. Toffoli. Pancake ice floe size distribution during the winter expansion of the Antarctic marginal ice zone. *The Cryosphere*, 13(1):41–48, 2019. doi: 10.5194/tc-13-41-2019.
- M. H. Derkani. *Waves in the Southern Ocean and Antarctic Marginal Ice Zone: Observations and modelling*. PhD thesis, 2021.
- O. Faltinsen. *Sea loads on ships and offshore structures*, volume 1. Cambridge university press, 1993.
- P. V. Guimarães, F. Ardhuin, F. Bergamasco, F. Leckler, J.-F. Filipot, J.-S. Shim, V. Dulov, and A. Benetazzo. A data set of sea surface stereo images to resolve space-time wave fields. *Scientific data*, 7(1):1–12, 2020. doi: 10.1038/s41597-020-0492-9.
- A. Kohout, M. Smith, L. A. Roach, G. Williams, F. Montiel, and M. J. M. Williams. Observations of exponential wave attenuation in Antarctic sea ice during the pipers campaign. *Annals of Glaciology*, page 1–14, 2020. doi: 10.1017/aog.2020.36.
- F. Montiel, A. L. Kohout, and L. A. Roach. Physical drivers of ocean wave attenuation in the marginal ice zone. *J. Phys. Oceanog.*, 2022. ISSN 0022-3670, 1520-0485. doi: 10.1175/JPO-D-21-0240.1.
- A. Womack, M. Vichi, A. Alberello, and A. Toffoli. Atmospheric drivers of a winter-to-spring lagrangian sea-ice drift in the eastern antarctic marginal ice zone. *Journal of Glaciology*, page 1–15, 2022. doi: 10.1017/jog.2022.14.
- World Meteorological Organization. *Guide to wave analysis and forecasting, WMO-No 702*. World Meteorological Organization Geneva, Switzerland, 1998. doi: <https://doi.org/10.25607/OBP-1523>.
- I.R. Young. On the measurement of directional wave spectra. *Applied Ocean Research*, 16(5):283–294, 1994. ISSN 0141-1187. doi: [https://doi.org/10.1016/0141-1187\(94\)90017-5](https://doi.org/10.1016/0141-1187(94)90017-5).

Response to Referee #2

We thank the Referee for the positive comments. Our comment-by-comment response follows.

This paper presents a novel dataset of waves propagating in the marginal ice zone of the Southern Ocean during an explosive cyclone. I see two main points that this paper makes. The first one is the demonstration that ship-based photogrammetry, and stereoscopy in particular, is a promising method for characterizing waves propagating in an ice cover. Indeed the method successfully measures very large waves quite accurately, with respect to independent measurements obtained with wave buoys, and it is described such that it can be implemented elsewhere. It is worth noting that the WASS is an open source software that has been used in many other applications. The second point is that attenuation rates comply with what's been measured and empirically modeled in previous studies, and that wind can be responsible for wave growth in 100% ice concentration. The latter is based on wave attenuation coefficients that were measured with wave buoys during another experiment in the Antarctic MIZ, and on a number of assumptions about how wave attenuation is calculated in this paper and how wind-wave generation occur. Those two results are relatively well conveyed and deserve to be published.

I see however some weaknesses in the paper that can be improved before publication. The comment can be summarized like this:

We have divided the comment and response into two parts.

wave-ice interactions are so complex, non-linear and variable that concluding about what's going on in a particular situation necessitates taking into account the possible errors that can influence the outcome. First, there is very little information about sea ice along the transects, even though an imaging system was used to characterize waves-in-ice. . . . Given that cameras were used, the results should mention how the ice properties evolved along the transect,

We note that Alberello et al. [2019] gave a detailed analysis of sea ice properties (floe diameters and size distribution) from the full collection of camera images. In response to this comment, we have added a new figure to the manuscript (Figure 2), showing the floe diameter and floe size distribution as a function of distance from the edge and corresponding photos. The results show that the average floe diameter slightly increases from ≈ 3 m at the edge to ≈ 3.5 m deeper in the marginal ice zone, but that the sea ice state was fairly homogeneous along the transect. The floe reconstruction technique is summarised in the Floe Size section of Methods.

The scientific literature arguing that thickness, floe size and other properties like ice rigidity affect wave attenuation is abundant, either theoretically and observationally. The discussion about wave attenuation suggests to the reader that the attenuation solely depends on ice concentration, and wave frequency. Stopa et al. (2018) (<https://doi.org/10.1073/pnas.1802011115>), using a very large data set obtained from satellite radar imagery, provide evidence that the attenuation rate in ice is highly variable and cover many orders of magnitude. . . . and the discussion should be adjusted at least to acknowledge the role of ice properties in setting the attenuation rate.

We have largely re-written the Introduction. In particular, in the third paragraph, we discuss the dependence of wave attenuation in the MIZ on sea ice properties, including references to Stopa et al. [2018] and the recent article by Montiel et al. [2022].

A second weakness lies in the calculation of the frequency wave spectra. Figures 2 and 3 do not show the noise level from the buoy, that is typically red, and from the stereo imaging. Typically also these figures are shown on a log scale. Thomson et al. [2021] describe a procedure that avoids contaminating attenuation results with noise. This should be discussed, especially when comparing data obtained from different sources (stereo and buoys).

We have added further analysis of the error associated to the image processing technique (WASS)

and report details in the last paragraph of the Estimation of the Surface Elevation section in Methods. The noise associated to the stereo images depends on the geometry of the installation and it is uniform across frequencies, i.e. white noise. In the figure below (not included in the manuscript), we summarise all measured spectra in logarithmic scale and we show the noise level i.e. noise integrated over frequency (shaded area). The figure highlights that noise only affects high frequency modes ($f > 0.3$ Hz or periods < 3.5 s), which contribute marginally to the calculation of integrated parameters. For example, the effect of white noise on the significant wave height is 0.5% and, thus, it is considered to be negligible. Similarly, the red noise in the buoys does not produce any notable effect on the significant wave height as reported in Thomson et al. [2021]. A brief discussion on the effect of noise is reported in the Three-dimensional Imaging of Ocean Surface and Comparison with Buoy Data section. Further, we have revised Figure 3 (previously Figure 2) to show wave spectra in both linear and logarithmic scales; an error band is shown for the spectra extracted from the stereo images to highlight uncertainty related to noise.

The procedure discussed in Thomson et al. [2021] to avoid contamination of noise bias on wave attenuation is only relevant for a frequency-dependent attenuation rate. Due to the non-stationarity of the sea state, a frequency-dependent analysis of wave attenuation is not performed with our measurements. Although we comment on the noise bias in the buoy (with reference to Thomson et al. [2021]; see Three-dimensional Imaging of Ocean Surface and Comparison with Buoy Data), specific details as to how noise contamination can be avoided are not reported in the revised manuscript.

Fig: Wave spectra in logarithmic scale with noise level from the image processing (white noise) highlighted by the grey band.

The footprint of the camera system is approximately 200m. What happens for wave component having a wavelength larger than this? Measuring the amplitude of such of wave would require a good knowledge of the absolute height, which is not straightforward using an IMU. Could the authors comment on this? Does that explain why the energy for long waves is less when measure with the camera than with the buoy (Fig. 2)?

The analysis is performed on time series extrapolated from virtual arrays within the field of view. The use of time series allows estimation of wave components longer than 200 m and, thus, a better definition of the wave spectrum. We clarified this in the Frequency–Direction Spectrum of Methods section.

The directionality of the wave spectra is another source of uncertainty that should be taken into account. In the current discussion, the energy is partitioned following the local wind, if I understand the algorithm correctly. In order to compute an attenuation coefficient between two stations, waves reaching the innermost station must have passed by the outermost station. Can the adequate par-

titioning target the direction connecting the two stations rather than the direction of the local wind seas, which according to Fig. 1 seems to be turning quite significantly?

Wind and wave directions were approximately aligned during our measurements, with a difference of $\approx 10^\circ$ according to the ERA5 reanalysis. Moreover, directions did not vary to a great extent throughout the experiment, with the wave propagation angle shifting from 319° at 8:00UTC to 312° at 12:00UTC. The swell system was offset by less than 20° degrees relative to wind sea. The temporal variation of wind, total sea and swell directions at the ice edge (in open ocean) is summarised in the figure below (not reported in the manuscript). Details on wind and wave directions have been added to the Experimental Conditions section.

Fig: Direction of total sea, swell, and wind at the sea ice edge (in open water) from ERA5 reanalysis. Convention coming from.

However, as noted by the Referee uncertainties in directions translate into inaccuracy in the calculation of the distance from the ice edge and, hence, the wave attenuation rate. To highlight errors, uncertainties in the distance from the ice edge have been reported in Figure 5 (Figure 4 in the original manuscript). As angular differences and offsets are only marginal, uncertainties are small and $\approx \pm 5\%$ at the deepest locations in the MIZ. This errors do not produce any effect in the overall trend of significant wave height. Additional details on these uncertainties have been added in the Estimation of the Attenuation Rate section of Methods and discussed in the end of the Wave Evolution section.

Since the conclusion about wind-generation depends on Fig. 4, which do not show uncertainties and which is still quite speculative, I strongly recommend that the various sources of uncertainties be better quantified, illustrated and discussed. Otherwise, the paper is very well written and I fully recognize the importance of this work, as well as the challenges that analyzing these data poses.

We updated the figure (Figure 5 in the revised manuscript) to account for uncertainties.

References

- A. Alberello, M. Onorato, L. Bennetts, M. Vichi, C. Eayrs, K. MacHutchon, and A. Toffoli. Pancake ice floe size distribution during the winter expansion of the Antarctic marginal ice zone. *The Cryosphere*, 13(1):41–48, 2019. doi: 10.5194/tc-13-41-2019.
- F. Montiel, A. L. Kohout, and L. A. Roach. Physical drivers of ocean wave attenuation in the marginal ice zone. *J. Phys. Oceanog.*, 2022. ISSN 0022-3670, 1520-0485. doi: 10.1175/JPO-D-21-0240.1.
- J. E. Stopa, P. Sutherland, and F. Ardhuin. Strong and highly variable push of ocean waves on Southern Ocean sea ice. *Proceedings of the National Academy of Sciences*, 115(23):5861–5865, 2018. ISSN 0027-8424. doi: 10.1073/pnas.1802011115.

J. Thomson, L. Hošeková, M. H. Meylan, A. L. Kohout, and N. Kumar. Spurious rollover of wave attenuation rates in sea ice caused by noise in field measurements. *Journal of Geophysical Research: Oceans*, 126(3):e2020JC016606, 2021. doi: <https://doi.org/10.1029/2020JC016606>.

REVIEWERS' COMMENTS

Reviewer #1 (Remarks to the Author):

I am satisfied with the responses and the paper is suitable for publication.